# IL-33 promotes innate lymphoid cell-dependent IFN-γ production required for innate immunity to *Toxoplasma gondii*

**Joseph T Clark[1], David A Christian[1], Jodi A Gullicksrud[1], Joseph A Perry[1], Jeongho Park[1,2], Maxime Jacquet[1,3], James C Tarrant[1], Enrico Radaelli[1], Jonathan Silver[4], Christopher A Hunter[1]\***

[1]Department of Pathobiology, University of Pennsylvania School of Veterinary Medicine, Philadelphia, United States; [2]Kangwon National University College of Veterinary Medicine and Institute of Veterinary Science, Chuncheon, Republic of Korea; [3]Liver Immunology, Department of Biomedicine, University Hospital of Basel and University of Basel, Basel, Switzerland; [4]Department of Respiratory Inflammation and Autoimmunity, AstraZeneca, Gaithersburg, United States

**Abstract** IL-33 is an alarmin required for resistance to the parasite *Toxoplasma gondii*, but its role in innate resistance to this organism is unclear. Infection with *T. gondii* promotes increased stromal cell expression of IL-33, and levels of parasite replication correlate with release of IL-33 in affected tissues. In response to infection, a subset of innate lymphoid cells (ILC) emerges composed of IL-33R+ NK cells and ILC1s. In *Rag1*−/−mice, where NK cells and ILC1 production of IFN-γ mediate innate resistance to *T. gondii*, the loss of the IL-33R resulted in reduced ILC responses and increased parasite replication. Furthermore, administration of IL-33 to *Rag1*−/− mice resulted in a marked decrease in parasite burden, increased production of IFN-γ, and the recruitment and expansion of inflammatory monocytes associated with parasite control. These protective effects of exogenous IL-33 were dependent on endogenous IL-12p40 and the ability of IL-33 to enhance ILC production of IFN-γ. These results highlight that IL-33 synergizes with IL-12 to promote ILC-mediated resistance to *T. gondii*.

**\*For correspondence:**
chunter@upenn.edu

## Introduction

*Toxoplasma gondii* is an intracellular parasite of public health significance (*Montoya and Liesenfeld, 2004*; *Weiss and Dubey, 2009*; *Israelski and Remington, 1988*). Resistance to this organism is initiated by dendritic cell production of IL-12, which promotes natural killer (NK) and T cell secretion of interferon-γ (IFN-γ) (*Khan et al., 1994*; *Hunter et al., 1994*; *Yap and Sher, 1999a*). IFN-γ in turn induces multiple anti-microbial mechanisms, which include the activation of macrophages to express inducible nitric oxide synthase (iNOS), which are required to limit parasite replication (*Yap and Sher, 1999b*; *Dunay et al., 2010*; *Serbina et al., 2003*; *Scharton-Kersten et al., 1997*; *Wang et al., 2019*; *Chen et al., 2020*). Previous studies have shown that mice deficient in the adapter molecule MyD88 have increased susceptibility to *T. gondii* associated with reduced production of IL-12 and IFN-γ (*Scanga et al., 2002*; *Del Rio et al., 2004*). Since MyD88 is a major adapter required for Toll-like receptor (TLR) signaling, this increased susceptibility is consistent with a role for TLR-mediated recognition of this pathogen. However, TLR1, TLR2, TLR4, TLR6, TLR9, and TLR11 are individually not required for early resistance to *T. gondii* (*Yarovinsky et al., 2005*; *Plattner et al., 2008*; *Hitziger et al., 2005*; *Furuta et al., 2006*), and there is a MyD88-independent mechanism of parasite recognition (*Kim et al., 2006*; *Sukhumavasi et al., 2008*; *Mercer et al., 2020*). Moreover, administration of IL-12 to *Myd88*−/− mice does not restore the ability to produce IFN-γ, and NK and

T cell expression of MyD88 is required for optimal production of IFN-γ and resistance to *T. gondii* (*LaRosa et al., 2008*; *Ge et al., 2014*). Thus, MyD88 has a critical role in resistance to *T. gondii,* but the events that engage this adapter molecule are unclear.

Members of the IL-1 family of cytokines, including IL-1α/β, IL-18, and IL-33, utilize distinct receptor sub-units but share downstream signaling machinery that includes MyD88. These cytokines impact a wide range of immune cells and influence many facets of the innate immune system (*Dinarello, 2018*). Mice that lack T and B cells have helped define the impact of cytokines on innate mechanisms of immunity to a wide variety of pathogens (*Powell et al., 2012*; *Monticelli et al., 2011*; *Klose and Artis, 2016*; *Abt et al., 2015*). For example, these models were important to identify the role of IL-12 in promoting NK cell production of IFN-γ required for innate resistance to *Listeria monocytogenes* and *T. gondii* (*Bancroft et al., 1991*; *Tripp et al., 1993*; *Hunter et al., 1993*). It is now appreciated that NK cells and ILC1 populations are both relevant sources of IFN-γ that contribute to the control of *T. gondii* (*Park et al., 2019*; *Weizman et al., 2017*; *López-Yglesias et al., 2020*). Although IL-1 and IL-18 synergize with IL-12 to promote NK cell production of IFN-γ (*Hunter et al., 1995a*; *Cai et al., 2000*; *Kearley et al., 2015*), the role of endogenous IL-1 during toxoplasmosis is secondary to those of IL-12 (*Hitziger et al., 2005*; *Hunter et al., 1995a*; *Melchor et al., 2020*), and endogenous IL-18 is not required for parasite control but rather contributes to the immune pathology that can accompany this infection (*Cai et al., 2000*; *Yap et al., 2001*; *Vossenkämper et al., 2004*; *Ewald et al., 2014*; *Gorfu et al., 2014*). In contrast, endogenous IL-33 has an essential role in resistance to *T. gondii*, but the basis for this protective effect is not well understood (*Jones et al., 2010*).

IL-33 is a cytokine that is constitutively expressed by endothelial and epithelial cells, and in current models, the death of these cells leads to the release of IL-33 that acts as a damage-associated molecular pattern (DAMP) or alarmin to activate immune cell populations (*Schmitz et al., 2005*; *Moussion et al., 2008*; *Liew et al., 2016*; *Rostan et al., 2015*). While there are open questions about whether this cytokine can also be secreted (*Kouzaki et al., 2011*; *Kakkar et al., 2012*), the rapid oxidation of IL-33 inactivates this cytokine and ensures that its activity is restricted to local sites of tissue damage (*Cohen et al., 2015*). IL-33 and the IL-33R (also termed ST2, encoded by *Il1rl1*) are most prominently associated with amplification of TH2 CD4[+] T cells, activation of ILC2, and resistance to helminths (*Schmitz et al., 2005*; *Humphreys et al., 2008*; *Silver et al., 2016*; *Osbourn et al., 2017*; *Ricardo-Gonzalez et al., 2018*; *Molofsky et al., 2015*), immune regulation by Treg cells (*Matta et al., 2014*; *Schiering et al., 2014*), and a number of metabolic and para-immune functions mediated by ILC2 and regulatory T cells (*Spallanzani et al., 2019*; *Ito et al., 2019*). Consistent with its ability to promote TH2-type responses, IL-33 can antagonize inflammation mediated by TH1/TH17 cells during experimental allergic encephalomyelitis (EAE) (*Franca et al., 2016*; *Milovanovic et al., 2012*; *Xiao et al., 2018*) and suppresses pathological TH1 responses during visceral leishmaniasis (*Rostan et al., 2013*). However, during infection with LCMV or MCMV, IL-33 promotes the expansion of NK cells and T cells and their production of IFN-γ, and loss of the IL-33R results in a delay in viral clearance (*Nabekura et al., 2015*; *Bonilla et al., 2012*; *Baumann et al., 2015*). In contrast, perhaps one of the most strking phenotypes of mice that lack IL-33R is that they succumb to chronic toxoplasmosis, associated with reduced astrocyte responses required for protective T cell responses (*Jones et al., 2010*; *Still et al., 2020*). Nevertheless, the acute stage of toxoplasmosis is associated with the ability of *T. gondii* to infect and lyse epithelial and endothelial cells (*Konradt et al., 2016*; *Van Grol et al., 2013*; *Delgado Betancourt et al., 2019*; *Luu et al., 2019*; *Ju et al., 2009*), but whether these events lead to the release of IL-33 or if this affects the innate response to Toxoplasma is unknown. The present study reveals that parasite replication during acute toxoplasmosis is associated with the release of IL-33, and *Rag1*[−/−] mice that lack the IL-33R have defects in ILC production of IFN-γ and impaired parasite control. Furthermore, administration of IL-33 to *Rag1*[−/−] mice enhanced ILC production of IFN-γ associated with the expansion of a population of Ly6c[hi] CCR2[+] inflammatory monocytes and a marked reduction in parasite burden. Together, these results highlight that infection-induced release of IL-33 synergizes with IL-12 to promote ILC production of IFN-γ required for resistance to *T. gondii*.

## Results

### *Toxoplasma gondii* infection induces IL-33 upregulation and release

To determine the impact of Toxoplasma infection on IL-33 expression and secretion, C57BL/6 WT and $Rag1^{-/-}$ mice were infected intraperitoneally (i.p.) with the Me49 strain or the replication-deficient CPS strain of *T. gondii*, and the levels of IL-33 at local sites of infection and affected tissues assessed by ELISA. In the peritoneum of naïve WT and $Rag1^{-/-}$ mice, the level of IL-33 was below the limit of detection (<10 pg/ml) (*Figure 1A*). Infection i.p. with $2 \times 10^5$ tachyzoites of the non-replicating CPS strain did not cause parasite-induced host cell lysis and failed to elicit detectable IL-33 at 1 or 5 days post-infection (dpi) (data not shown). Intraperitoneal infection of WT mice with 20 cysts of Me49 resulted in <1% infected cells in the peritoneum at 5 dpi, and IL-33 was not detected by ELISA (*Figure 1A*). When $Rag1^{-/-}$ mice received the same challenge, there were 2–5% infected cells at 5 dpi, and elevated levels of IL-33 were present (*Figure 1A*). To test whether IL-33 levels were a function of parasite burden, WT and $Rag1^{-/-}$ mice were treated with anti-IFN-γ, which resulted in a 20-fold increase in parasite load (data not shown) and a threefold to fourfold increase in the levels of IL-33 (*Figure 1A*). When these data sets were collated and quantity of parasite DNA plotted versus IL-33 concentration, there was a strong correlation between parasite burden and IL-33 levels (R = 0.7902) (*Figure 1B*). To determine whether IL-33 was released in other tissues affected by *T. gondii*, tissue biopsies from the liver of WT mice at 10 dpi were prepared and placed in culture for 24 hr and IL-33 release measured. While basal levels of IL-33 were detected in tissues from naïve WT mice and mice injected with replication-deficient CPS parasites, the biopsies from infected mice showed significantly elevated levels of IL-33 (*Figure 1C*). These results suggest that parasite replication and lysis of infected cells lead to IL-33 release, and these levels are comparable to those reported in other inflammatory settings (*Kearley et al., 2015*; *Llop-Guevara et al., 2014*).

To identify the cellular source of IL-33 during infection, the IL-33-IRES-GFP mouse (*Johnston et al., 2016*), a faithful reporter for IL-33 protein production in comparison to direct staining and $Il33^{-/-}$ mice (*Figure 1—figure supplement 1A*), was utilized. The IL-33 reporter mice were infected i.p. with a fluorescent strain of *T. gondii* (Pru-tdTom), and the expression of IL-33-GFP in the omentum was examined by flow cytometry and IHC at 3 dpi. The omentum is an adipose tissue that contains fat-associated lymphoid clusters (FALCs), which are one of the major sites for drainage from the peritoneum (*Christian et al., 2020*; *Jackson-Jones et al., 2016*; *Buscher et al., 2016*). In naïve mice, the omentum contained a small population of CD45$^+$ immune cells, most of which were IL-33-GFP$^-$, whereas fibroblastic stromal cells (CD45$^-$FSC$^{hi}$ SSC$^{hi}$ CD31$^{+/-}$ PDPN$^{+/-}$) were the main source of IL-33-GFP$^+$ cells (*Figure 1D*, *Figure 1—figure supplement 1B*). Upon infection, there was a marked (approximately 7- to 10-fold) expansion of the CD45$^+$ population and a small population (0.5%) of the CD45$^+$ F4/80$^+$ MHCII$^+$ cells expressed IL-33 (*Figure 1D* and data not shown). Nevertheless, although the percentage of stromal cells that expressed IL-33 was decreased (as a consequence of the increased inflammatory populations, compare cell density of plots in *Figure 1D*), the major population of IL-33-GFP$^+$ cells remained fibroblastic stromal cells (*Figure 1—figure supplement 1B*). The ability to detect infected cells based on parasite expression of tdTomato revealed that infected cells were not associated with IL-33 expression (*Figure 1—figure supplement 1C*), suggesting that infection of individual cells does not directly drive IL-33 expression. Similarly, at 7 dpi, the use of flow cytometry and immunofluorescence revealed that CD45$^-$ cells were the dominant source of IL-33 in the spleen, lung, and liver (*Figure 1—figure supplement 1D*). By contrast, the majority of IL-33R staining in infected mice was observed on CD45$^+$ cells, suggesting that hematopoietic cells are the primary responders to IL-33 release (*Figure 1—figure supplement 1E*). To understand the spatial organization of the IL-33-GFP$^+$ cells, the omentum was used for whole tissue mount immunofluorescence. In uninfected mice, consistent with the analysis above, IL-33 was constitutively expressed by non-hematopoietic CD45$^-$ cells with fibroblastic morphology distributed throughout the FALCs. At 3 dpi, there was a marked increase in the size of the FALC, and an approximate 3-fold increase in number of IL-33$^+$ cells (*Figure 1E*). These images are max projection views that illustrate the size of FALCs, but quantification of the intensity of fluorescence highlighted the 10-fold increase in the expression of IL-33-GFP (*Figure 1E*) associated with ERTR7$^+$ fibroblastic reticular cells (*Figure 1F*). Imaging revealed that areas of parasite replication were inversely correlated with the presence of IL-33-GFP expression (*Figure 1G*). Together, these data establish that in mice acutely infected with *T. gondii* stromal cells are a major source of IL-33, that release of IL-33

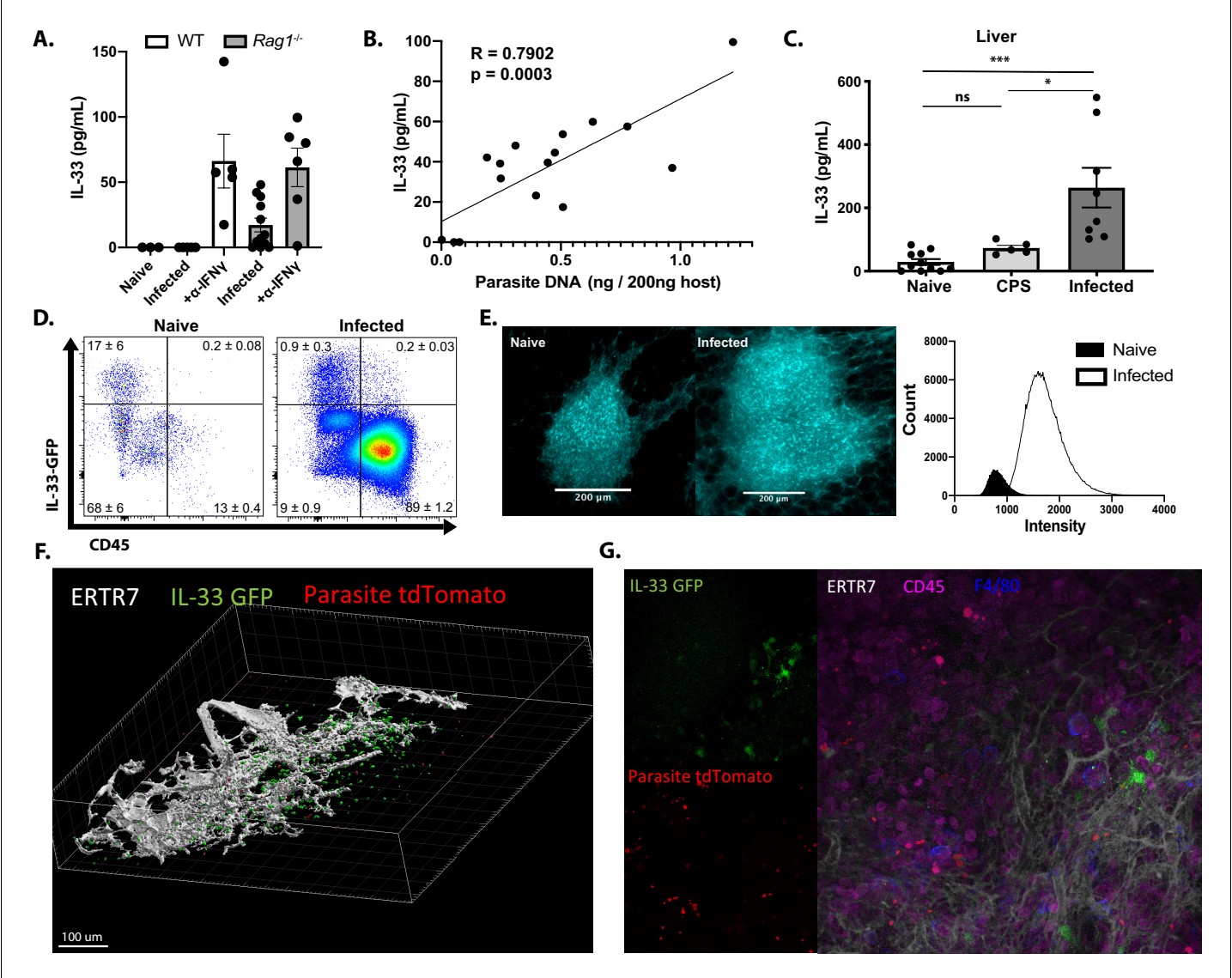

**Figure 1.** *Toxoplasma gondii* infection induces IL-33 expression and release. Mice were infected i.p. with *T. gondii*. After 7 days, (**A**) free IL-33 in the peritoneal cavity was measured by ELISA. (**B**) Measurements of IL-33 from (**A**) were plotted against corresponding parasite burden and fit to a linear model. (**C**) Five millimeter punch biopsies of liver was placed in culture for 24 hr and IL-33 measured in supernatants by ELISA. Each point represents the mean of three biopsies from a single mouse. (**D**) Cells from omenta of IL-33 GFP reporter mice were analyzed by flow cytometry at 3 days post-infection. Cells shown are live singlets. Data are representative of three mice per group. (**E**) Whole mount omentum showing IL-33-GFP signal in milky spot. (**F**) 3D projection of milky spot showing stromal marker ERTR7 and IL-33 GFP signal. (**G**) Whole mount immunofluorescence of milky spot. NS, not significant (p>0.05); *p<0.05 and ***p<0.001 (one-way ANOVA with Tukey's multiple comparisons test). Data are pooled from three (**A and B**) or two (**C, D, E, F, and G**) independent experiments (mean + s.e.m.).

The online version of this article includes the following source data and figure supplement(s) for figure 1:

**Source data 1.** Excel file containing numerical values collected from IL-33 ELISA and parasite DNA qPCR shown in *Figure 1*.

**Figure supplement 1.** IL-33 is primarily expressed by non-hematopoietic cells.

correlates with levels of parasite replication, and that a sub-population of innate immune cells express the IL-33R.

## ILC responses to IL-33

To identify the cell populations that could respond to the local release of IL-33 during this infection, a UMAP analysis was used to provide an unbiased comparison of the changes in IL-33R expression

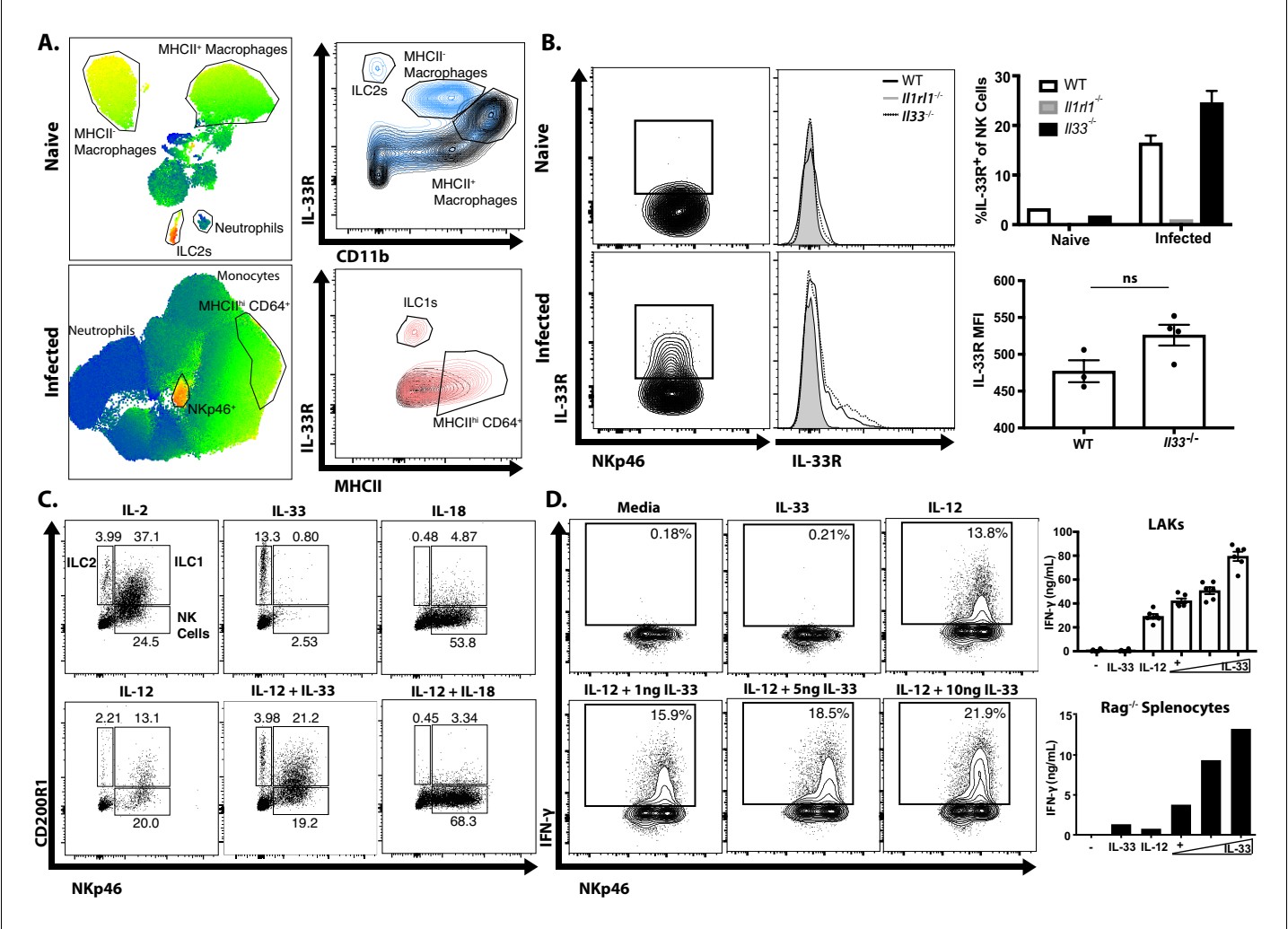

**Figure 2.** Infection sensitizes NK and ILC to IL-33. (A) UMAP analysis of peritoneal exudate cells from naïve or 7 dpi i.p. mice, with heatmap for IL-33R expression. Data compiled from four mice per group. (B) Flow cytometry from peritoneal cells showing IL-33R staining on NKp46+ cells. Data are representative of three to four mice per group. (C) Flow cytometry of LAKs showing composition of population based on cytokine stimulation condition. Population shown is pre-gated on live singlets. (D) Intracellular cytokine staining of LAKs after 24 hr cytokine stimulation and 4 hr incubation with Brefeldin A. NS, not significant (p>0.05) (Student's t-test); data are representative of three independent experiments (A–D).

The online version of this article includes the following source data for figure 2:

**Source data 1.** Excel file containing numerical values collected from IL-33R staining and IFN-γ ELISA shown in *Figure 2*.

in the peritoneum of naïve and infected *Rag1*$^{-/-}$ mice (*Figure 2A*). In naïve mice, IL-33R was expressed by peritoneal macrophages (CD64$^+$CD11b$^+$MHCII$^{+/-}$) and a small population of ILC2 (Lin$^-$ Nkp46$^-$) when compared with *Il1rl1*$^{-/-}$ controls (*Figure 2A*). By 5 dpi, there was a marked change in the cellular composition of the peritoneum with a loss of the MHCII$^-$ macrophage and ILC2 populations but a prominent monocyte and neutrophil infiltration and the expansion of NKp46$^+$ NK cells and ILC1s (*Figure 2A*, bottom). While there were low levels of IL-33R expressed by MHCII$^{hi}$ CD64$^+$ cells, the highest levels of IL-33R were observed on NKp46$^+$ cells. Comparison of WT, *Il1rl1*$^{-/-}$, and *Il33*$^{-/-}$ mice revealed that IL-33R expression was not detected on peritoneal or splenic NK cells in naïve mice, but IL-33R was observed on a subset (~20%) of NK cells in the perito- neum by 5 dpi (*Figure 2B*). Furthermore, NK cells from infected *Il33*$^{-/-}$ mice still upregulated expression of the IL-33R, indicating that IL-33 signaling is not required for this process. Thus, acute infection of immune competent mice with *T. gondii* is characterized by the emergence of popula- tions of NK cells and ILC1s that express the IL-33R.

To understand the impact of IL-33 on ILC populations, IL-2 induced lymphokine-activated killer cells (LAKs) generated from the bone marrow of *Rag1*⁻/⁻ mice (**Hunter et al., 1997**; **Wherry et al., 1991**) were utilized to compare the impact of IL-33 (and its relative IL-18) alone or in combination with IL-12 on ILCs. Phenotyping of these LAK cultures revealed that they contained ILC1s (NKp46⁺ CD200R1⁺), ILC2s (NKp46⁻ CD200R1⁺) and NK cells (NKp46⁺ CD200R1⁻) (**Figure 2C**). Upon withdrawal of IL-2, the addition of IL-33 preferentially stimulated the proliferation of CD200R1⁺ ILC2s, while IL-18 stimulated NK cell proliferation (**Figure 2C**). IL-12 alone did not induce the expansion of a specific cell type, but when combined with IL-33 maintained the heterogeneity of the LAK population, while IL-12 plus IL-18 resulted in a modest increase in the proportion of NK cells compared to IL-18 alone. Moreover, while IL-33 alone did not stimulate LAKs to produce IFN-γ it did synergize

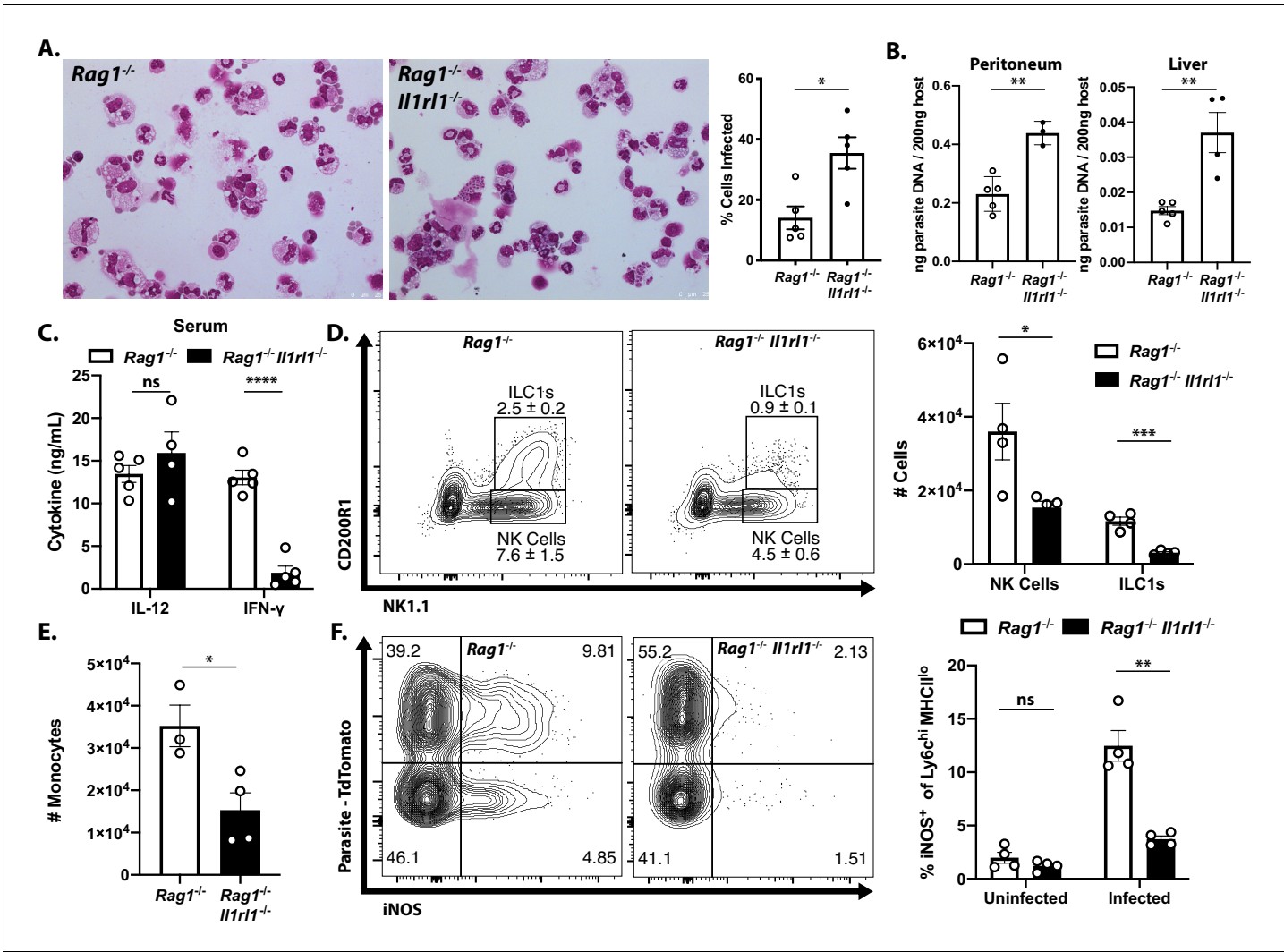

**Figure 3.** Endogenous IL-33 promotes the anti-parasitic immune response. (**A**) Cytospins of peritoneal exudate cells at 7 dpi i.p. (**B**) qPCR for parasite DNA from indicated tissues. (**C**) Serum cytokines measured by ELISA at 7 dpi. Representative of four to five mice per group. (**D**) Flow cytometric analysis and quantification of liver innate lymphoid cells. Populations shown are pre-gated on live singlets that are MHCII⁻. (**E**) Quantification of inflammatory monocytes (CD11b+CD64+Ly6 g−) in livers of infected mice at 7 dpi. (**F**) Intracellular iNOS staining from monocytes in (**E**), sub-gated on primary iNOS-producing cells (Ly6cʰⁱ MHCIIˡᵒ). NS, not significant (p>0.05); *p<0.05, **p<0.01, ***p<0.001, and ****p<0.0001 (student's t-test). Data are representative of three independent experiments.

The online version of this article includes the following source data and figure supplement(s) for figure 3:

**Source data 1.** Excel file containing numerical files collected from cytospin infected cell frequency quantification, parasite DNA qPCR, serum IL-12 and IFN-γ ELISA, and enumeration of cells shown in *Figure 3* and *Figure 3—figure supplement 1*.

**Figure supplement 1.** Endogenous IL-33 promotes anti-parasite responses in WT mice.

with IL-12 to enhance the production of IFN-γ in the NKp46$^{+}$ populations (*Figure 2D*). Similar results were observed when splenocytes from *Rag1*$^{-/-}$ mice were used (*Figure 2D*, bottom right panel) indicating that these effects of IL-33 on NK cells were not dependent on pre-activation with IL-2. These observations are consistent with previous reports on the ability of IL-33 to promote ILC2 activity (*Monticelli et al., 2011*; *Monticelli et al., 2015*), but demonstrate that in the presence of IL-12, IL-33 is a potent inducer of IFN-γ.

## Endogenous IL-33 is required for innate resistance to *T. gondii*

To directly test the role of endogenous IL-33 in innate resistance to *T. gondii*, *Rag1*$^{-/-}$ mice that lacked the IL-33R (*Rag1*$^{-/-}$; *Il1rl1*$^{-/-}$ mice) were generated and infected i.p. with *T. gondii*. Compared to *Rag1*$^{-/-}$ mice, at 7 dpi the *Rag1*$^{-/-}$; *Il1rl1*$^{-/-}$ mice showed an increased parasite burden based on the frequency of infected cells in the peritoneum (*Figure 3A*) and quantitation of parasite DNA in the peritoneum and liver (*Figure 3B*). Serum analysis of infected mice revealed comparable levels of IL-12p40 in *Rag1*$^{-/-}$ and *Rag1*$^{-/-}$; *Il1rl1*$^{-/-}$ mice, but IFN-γ was severely compromised in the *Rag1*$^{-/-}$; *Il1rl1*$^{-/-}$ mice (*Figure 3C*). Similar results were observed in *Rag1*-sufficient WT and *Il1rl1*$^{-/-}$ mice (*Figure 3—figure supplement 1A,B*). At this time point, *Rag1*$^{-/-}$ mice had a marked expansion in ILC1s and NK cells in the liver that was reduced in the absence of the IL-33R (*Figure 3D*). Consistent with decreased production of IFN-γ, fewer Ly6c$^{hi}$ monocytes were recruited to the liver in the *Rag1*$^{-/-}$; *Il1rl1*$^{-/-}$ mice (*Figure 3E*). Analysis of the Ly6c$^{hi}$ population in the liver at 7 dpi after infection with Pru-tdTom showed that a proportion of infected and uninfected cells express iNOS in the *Rag1*$^{-/-}$ mice, but iNOS levels were markedly reduced in the *Rag1*$^{-/-}$; *Il1rl1*$^{-/-}$ mice (*Figure 3F*). Neutrophil numbers, by contrast, were elevated in *Rag1*$^{-/-}$; *Il1rl1*$^{-/-}$ mice (*Figure 3—figure supplement 1C*), although this was not sufficient for parasite control. These data sets establish that endogenous IL-33 is required for optimal production of innate IFN-γ and the recruitment of monocyte populations that express anti-microbial effector mechanisms required for resistance to *T. gondii*.

## IL-33 treatment boosts IL-12- and IFN-γ-dependent immunity

Based on the ability of IL-33 to stimulate IL-12-dependent IFN-γ production in ILC1s and NK cells, a recombinant version of IL-33, resistant to oxidation which has a 30-fold increase in efficacy (*Kearley et al., 2015*; *Cohen et al., 2015*), was utilized to determine if exogenous IL-33 could be used to enhance innate resistance to *T. gondii*. Beginning at 1 dpi, IL-33 was administered i.p. every 2 days until 7 dpi, which resulted in a dose-dependent reduction in the frequency of infected cells at the site of infection and a decrease in parasite DNA in multiple tissues (*Figure 4A*). Analysis of cytospins of PECs revealed that treatment of infected *Rag1*$^{-/-}$ mice with IL-33 resulted in the emergence of a highly activated monocyte population (*Figure 4B*). By contrast, neutrophil numbers were not affected by IL-33 treatment (data not shown). These inflammatory monocytes were larger (higher FSC) and more granular (higher SSC) (*Figure 4C*). Furthermore, these cells were characterized by their expression of CD11b, CD11c, Ly6c, CCR2, and MHCII (*Figure 4C*). IL-33 treatment also resulted in increased recruitment of Ly6c$^{hi}$ CCR2$^{+}$ inflammatory monocytes to the liver and lungs by 7 dpi, and these monocytes had enhanced iNOS and IL-33R expression (*Figure 4D*). Histological analysis of the liver confirmed that IL-33 treatment resulted in increased cellular infiltration and expression of iNOS (*Figure 4E*, black arrows). Importantly, these changes induced by IL-33 treatment were associated with decreased necrotic foci that were frequent in infected *Rag1*$^{-/-}$ mice (*Figure 4E*, blue arrow). These results correlate the protective effects of IL-33 treatment with an increase in macrophage and monocyte responses required for the control of *T. gondii*.

To determine whether the protective effects of exogenous IL-33 depended on the ability of IL-12 to promote ILC production of IFN-γ, infected *Rag1*$^{-/-}$ mice were treated with IL-33 in combination with either anti-IL-12p40 or anti-IFN-γ neutralizing antibodies. Additionally, *Rag2*$^{-/-}$; *Il2rg*$^{-/-}$ mice, which lack ILCs, were treated with PBS or IL-33. Blockade of either IL-12 or IFN-γ entirely abrogated the protective effects of IL-33 treatment as measured by the frequency of infected cells in the peritoneum at 7 dpi (*Figure 5A*). As expected, *Rag2*$^{-/-}$; *Il2rg*$^{-/-}$ mice were more susceptible than *Rag1*$^{-/-}$ mice, and IL-33 treatment did not affect parasite burden in the peritoneum. IFN-γ levels at the site of infection were increased by IL-33 treatment in ILC-sufficient *Rag1*$^{-/-}$ animals, but were unaffected in *Rag2*$^{-/-}$; *Il2rg*$^{-/-}$ animals (*Figure 5B*). In *Rag1*$^{-/-}$ mice, treatment with IL-33 resulted in an

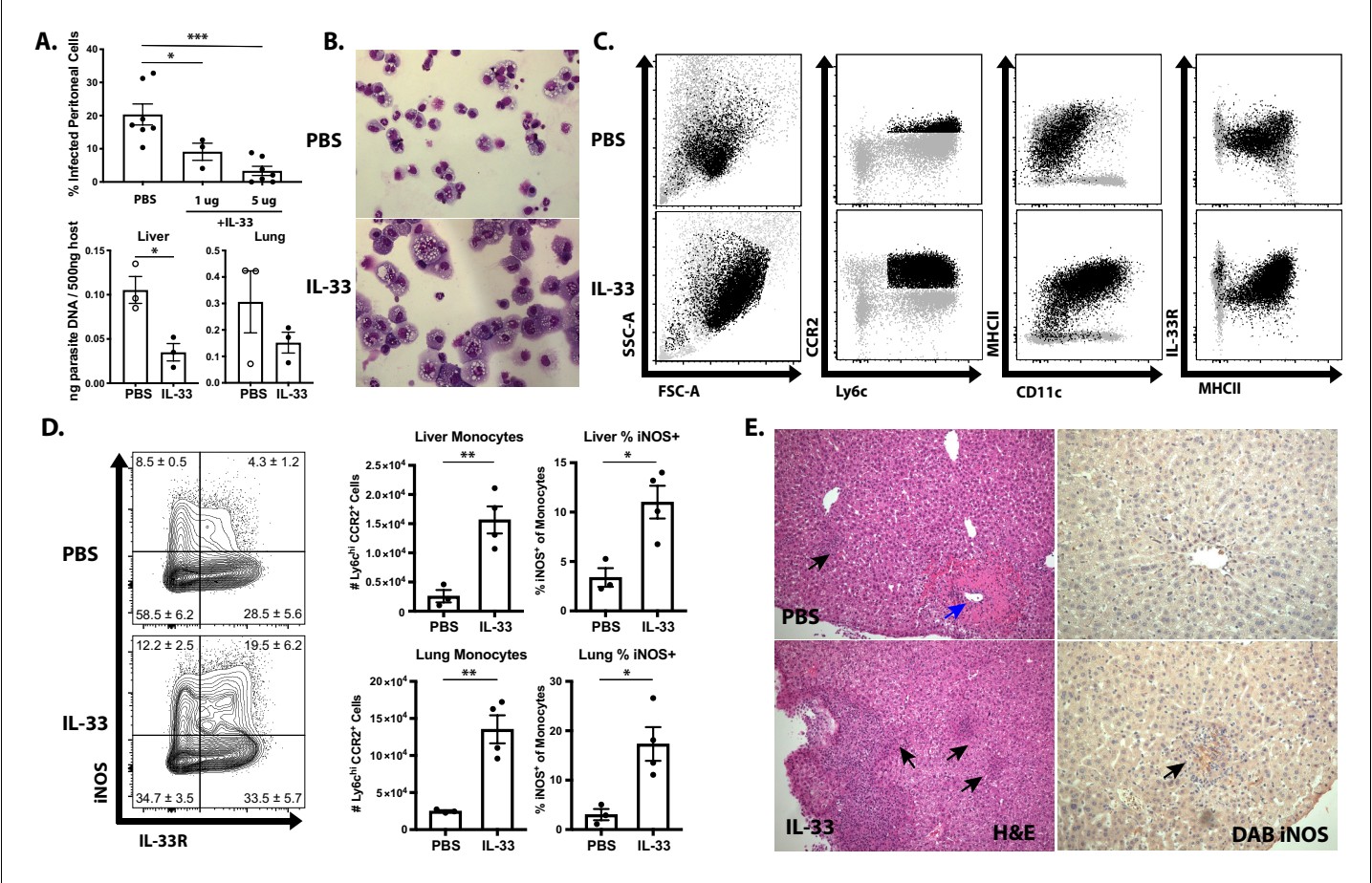

**Figure 4.** IL-33 treatment boosts IL-12- and IFN-γ-dependent immunity. (A) Quantification of infected cell frequencies in cytospins at 7 dpi i.p. and qPCR for parasite DNA in indicated tissues. (B) Representative cytospins from peritoneal lavage at 7 dpi. Data are representative of four to six mice per group. (C) Flow cytometric analysis of inflammatory monocytes in the peritoneal exudate at 7 dpi. Populations shown are pre-gated on live Ly6g⁻ singlets. Ly6c^hi CCR2⁺ cells are highlighted in black. (D) Representative analysis of Ly6c⁺ CCR2⁺ cells at 7 dpi in the liver and quantification of monocyte numbers and iNOS staining. (E) Histology of liver at 7 dpi, H and E showing infiltration of immune cells (left) and DAB iNOS staining (right). Black arrows indicate inflammatory infiltration; blue arrow indicates necrotic lesion. *p<0.05, **p<0.01, and ***p<0.001 (Student's t-test). Data are representative of five (A–C) or three (D–E) independent experiments.

The online version of this article includes the following source data for figure 4:

**Source data 1.** Excel file containing numerical values collected from cytospin infected cell frequency quantification, cell counts, and iNOS staining shown in *Figure 4*.

expansion of the NK and ILC1 compartments and their production of IFN-γ (*Figure 5C*). The expansion of Ly6c^hi CCR2⁺ monocytes associated with protection was also dependent on these factors, as cytokine blockade or absence of innate lymphoid cells resulted in the loss of these cells (*Figure 5D* and *Figure 5E*). These results suggest that the protective effects of IL-33 are not working through effects on the monocytes but rather that the ability of exogenous IL-33 to promote parasite control are dependent on IL-12 and ILC production of IFN-γ and subsequent activation of inflammatory monocytes.

## Discussion

Previous studies have identified a central role for IL-12 in innate and adaptive production of IFN-γ required for control of *T. gondii* (*Khan et al., 1994*; *Hunter et al., 1994*; *Gazzinelli et al., 1993*; *Hunter et al., 1995b*; *Yap et al., 2000*; *Wilson et al., 2008*), but other cytokines and costimulatory pathways potentiate the effects of IL-12 on NK cells. In particular, IL-1 and IL-18 can amplify NK cell

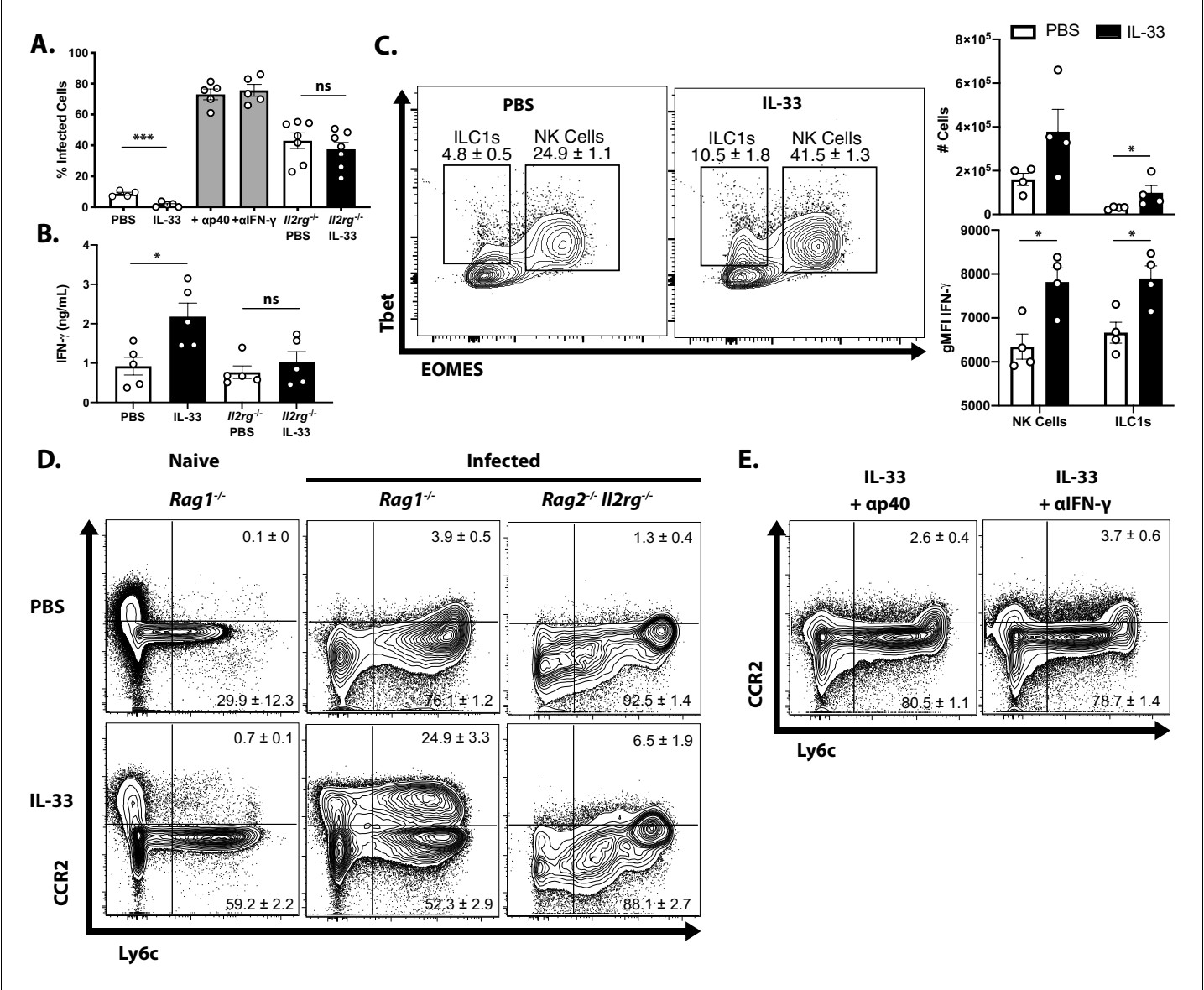

**Figure 5.** Protective effect of IL-33 is dependent on IL-12, IFN-γ, and ILC. (A) Quantification of cytospins from peritoneal exudate cells at 7 dpi i.p. (B) Quantification of IFN-γ in peritoneal lavage at 7 dpi. (C) Representative flow cytometric analysis of NK cells and ILC1s (left) and quantification of cell numbers and cytokine production (right) in the peritoneal exudate at 7 dpi. Population shown is pre-gated on live Ly6g⁻ NKp46⁺ singlets. (D and E) Flow cytometric analysis of inflammatory monocytes in peritoneum at 7 dpi. Data are representative of two independent experiments.

The online version of this article includes the following source data for figure 5:

**Source data 1.** Excel file containing numerical values collected from cytospin infected cell frequency quantification, IFN-γ ELISA, and enumeration of cells shown in *Figure 5*.

production of IFN-γ, but evidence that endogenous IL-1 and IL-18 are critical for control of *T. gondii* in this model is limited. Thus, while IL-1 or IL-18 contribute to the development of infection-induced, microbiome-dependent immune mediated pathology in the gut, there is limited evidence that loss of IL-1 or IL-18 leads to increased parasite replication (*Hitziger et al., 2005*; *LaRosa et al., 2008*; *Melchor et al., 2020*; *Vossenkämper et al., 2004*; *Mordue et al., 2001*; *Struck et al., 2012*; *Muñoz et al., 2015*; *Villeret et al., 2013*). Indeed, early studies showed that neutralization of endogenous IL-18 did not affect levels of parasite replication and that in SCID mice treated with IL-12 the effects of IL-1R blockade were modest and these treated mice were still more resistant than untreated SCID mice (*Hunter et al., 1995a*). It is relevant to note that recent work has highlighted a

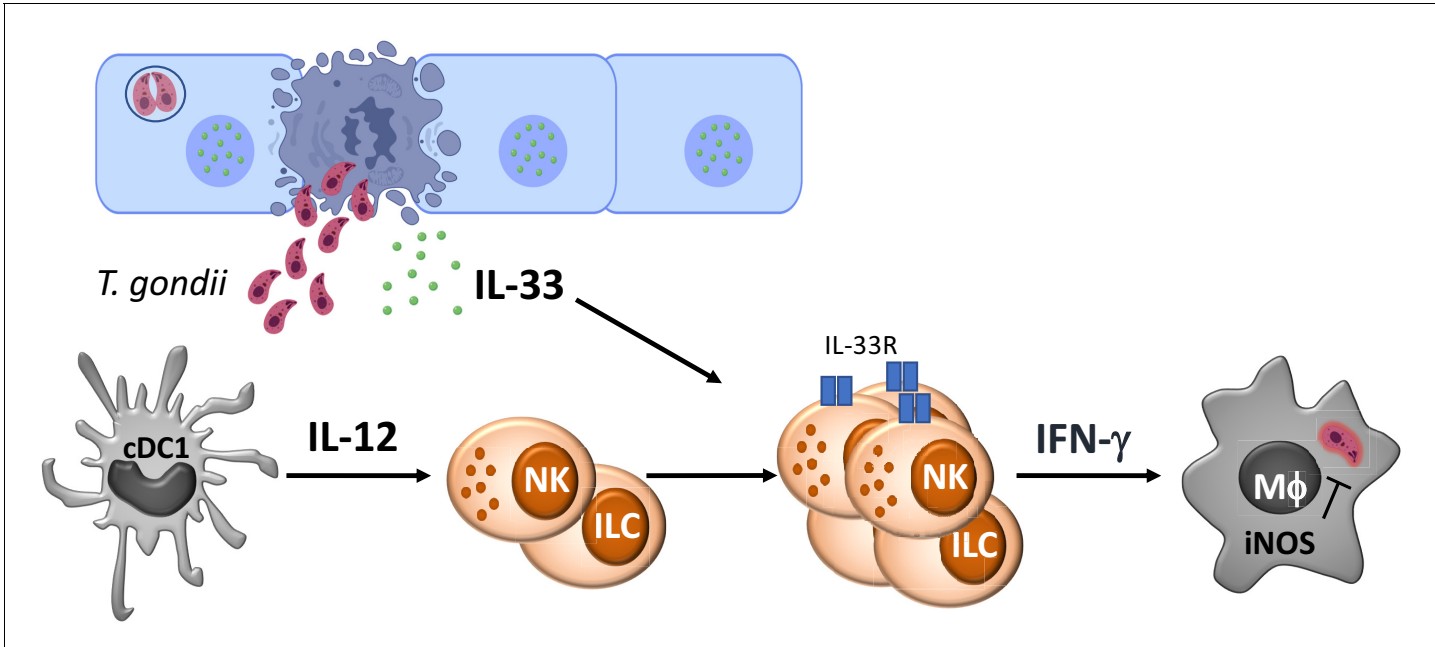

**Figure 6.** Model for the role of IL-33 in innate immunity to *Toxoplasma gondii*.

cell intrinsic role for MyD88 in NK cells to help control *T. gondii* (*Ge et al., 2014*), but neither IL-1 blockade or the use of IL-1R$^{-/-}$ and IL-18$^{-/-}$ mice replicates the susceptibility of *Myd88*$^{-/-}$ mice (*LaRosa et al., 2008*; *Hitziger et al., 2005*). In the studies presented here, the reduced NK and ILC responses observed in *Rag1*$^{-/-}$; *Il1rl1*$^{-/-}$ mice suggest that the ability of IL-33 (rather than IL-1 or IL-18) to amplify the IL-12-mediated innate response to acute toxoplasmosis helps explain the role for MyD88 in innate resistance to *T. gondii*. It is increasingly appreciated that in addition to NK cells, tissue resident ILC1s are an early source of IFN-γ in the innate response to *T. gondii* (*Park et al., 2019*; *Weizman et al., 2017*), but the differential programming of these cells, including their responsiveness to the IL-1 family member cytokines, is still being described. While IL-12 is central to NK and ILC1 production of IFN-γ, there are other stimuli that can potentiate this pathway (*Walker et al., 1999*). Certainly in vitro, in the presence of IL-12, IL-33 can promote NK cells and ILC1 IFN-γ production. While IL-18 is a more potent stimulator of IFN-γ in LAK cultures, the ability of IL-33 to promote NK and ILC1 responses, combined with the defects seen in these populations in *Il1rl1*$^{-/-}$ mice, suggests that of all the IL-1 family members, IL-33 is uniquely important for innate lymphoid cell responses to Toxoplasma.

Because the signaling pathways for the IL-1 family members converge on MyD88-dependent activation of NF-κB, differences in expression patterns and tissue localization are likely to dictate the relative importance of each cytokine. The release of IL-1 and IL-18 is typically considered to be downstream of inflammasome-mediated caspase activation and processing of pro-forms of these cytokines (*Man and Kanneganti, 2015*). While there is evidence that pro-IL-1 and pro-IL-18 are produced during toxoplasmosis (*Hunter et al., 1993*; *Zediak and Hunter, 2003*), several studies have concluded that *T. gondii* does not readily activate inflammasomes and there is evidence that Toxoplasma suppresses inflammasome activity (*Ewald et al., 2014*; *Lima et al., 2018*; *Liu et al., 2019*). However, there is a report that the inflammasome sensors NLRP1 and NLRP3 are required for protective immunity to *T. gondii* (*Gorfu et al., 2014*). A possible explanation for this discrepancy is that some inflammasome components are not just microbial sensors but have additional functions that include a role for Caspase eight in the activation of the c-Rel transcription factor required for expression of IL-12 and resistance to *T. gondii* (*DeLaney et al., 2019*). To date, no murine sensor of Toxoplasma or parasite ligand has been identified that directly activates inflammasomes, although there is evidence for sensor-independent routes for inflammasome activation (*Sandstrom et al., 2019*). In contrast to the complex events that lead to the production and processing of IL-1 and IL-18, IL-33 is expressed constitutively by epithelial and endothelial cells at barrier sites and stored in the nucleus,

and may therefore be resistant to parasite mechanisms of immune evasion and suppression that target host cell transcription. The release of IL-33 can occur as a consequence of tissue damage associated with allergic inflammation or viral infection (*Bonilla et al., 2012*; *Silver et al., 2016*), and during toxoplasmosis, IL-33 levels correlated closely with levels of parasite replication and host cell lysis. Thus, even though there may be non-canonical pathways for IL-33 release (*Kouzaki et al., 2011*; *Kakkar et al., 2012*; *Spallanzani et al., 2019*; *Snelgrove et al., 2014*; *Chen et al., 2015*), it seems likely that these physiological levels of IL-33 are a consequence of parasite-mediated lysis of infected cells.

Treatment of infected mice with exogenous IL-33 confirmed the protective effects of IL-33 and highlighted the impact on the recruitment of inflammatory monocytes to sites of infection and the subsequent upregulation of iNOS, a process required for the control of *T. gondii* (*Yap and Sher, 1999b*; *Dunay et al., 2010*; *Serbina et al., 2003*; *Scharton-Kersten et al., 1997*). IL-33 drives ILC-cell-dependent recruitment of CCR2$^+$ inflammatory monocytes, which resemble the TipDCs (TNF and iNOS-producing dendritic cells) previously shown to be important for control of infection (*Schiering et al., 2014*; *Spallanzani et al., 2019*). This finding is similar to an allergy model in which IL-33 contributes to the CCR2-dependent recruitment of inflammatory monocytes (*Tashiro et al., 2016*). It has been reported that IL-33 can directly enhance monocyte production of iNOS (*Li et al., 2014*), but in the studies reported here, the ability of exogenous IL-33 to promote this population of IL-33R$^+$ monocytes was dependent on ILC and the production of IFN-γ suggests that IL-33 is not sufficient to expand these monocytes. Additional experiments will be required to directly assess the contribution of the IL-33R to the regulation of monocyte function. It is also important to note that endogenous IL-33 is susceptible to rapid inactivation via oxidation in the extracellular space, which restricts its effects spatially and temporally. However, the recombinant IL-33 used in these studies was engineered to resist oxidation and it is possible that this treatment approach may have wider activities on hematopoiesis than IL-33 produced at sites of inflammation.

While IL-33 is most prominently linked to the regulation of Th2 type responses, there are reports that highlight the context dependent role that IL-33 plays in TH1 responses. In models of Leishmaniasis and cerebral malaria, IL-33 contributes to T-cell-dependent immune pathology in the skin and brain, respectively (*Rostan et al., 2013*; *Reverchon et al., 2017*). However, with the viral pathogens MCMV and LCMV, IL-33 contributes to NK and T cell expansion, and in its absence, there is a delay in viral clearance, but in neither case, is IL-33 essential for protective immunity (*Kearley et al., 2015*; *Nabekura et al., 2015*). Indeed, during intracerebral LCMV infection IL-33 contributes to the development of lethal immune pathology (*Bonilla et al., 2012*), whereas for mice chronically infected with *T. gondii*, the loss of IL-33 results in increased parasite burden (*Jones et al., 2010*). More recent studies have highlighted that IL-33 promotes astrocyte responses required for control of *T. gondii* in the CNS (*Still et al., 2020*). Nevertheless, the data presented here establish that the ability of IL-33 to amplify ILC responses and their production of IFN-γ plays a protective role in the acute innate response to Toxoplasma (see *Figure 6*). These results are consistent with a model in which IL-33 has a protective rather than pathological role in the immune response to *T. gondii*.

## Materials and methods

**Key resources table**

| Reagent type (species) or resource | Designation | Source or reference | Identifiers | Additional information |
|---|---|---|---|---|
| Gene (*Mus musculus*) | *Il33* | GenBank | MGI:1924375 | https://www.ncbi.nlm.nih.gov/gene/77215 |
| Gene (*Mus musculus*) | *Il1rl1* | GenBank | MGI:98427 | https://www.ncbi.nlm.nih.gov/gene/17082 |
| Strain, strain background (*Mus musculus*) | C57BL/6NTac | Taconic | RRID:MGI:5658006 | |
| Genetic reagent (*Mus musculus*) | B6.129S7-*Rag1*$^{tm1Mom}$/J | Jackson | RRID:IMSR_JAX:002216 | |

*Continued on next page*

Continued

| Reagent type (species) or resource | Designation | Source or reference | Identifiers | Additional information |
|---|---|---|---|---|
| Genetic reagent (*Mus musculus*) | C57BL/6NTac.*Rag2*tm1Fwa;*Il2rg*tm1Wjl | Taconic | Cat # 4111 | |
| Genetic reagent (*Mus musculus*) | B6(129S4)-Il33tm1.1Bryc/J | Jackson | RRID:IMSR_JAX:030619 | |
| Genetic reagent (*Mus musculus*) | *Il1rl1*tm1Anjm | PMID:10727469 **Townsend et al., 2000** | MGI:2386675 | http://www.informatics. jax.org/allele/MGI:2386675 |
| Strain, strain background (*Toxoplasma gondii*) | ME49 | NCBI:txid508771 | | |
| Strain, strain background (*Toxoplasma gondii*) | Pru-tdTomato | PMID:19578440 **John et al., 2009** | | |
| Strain, strain background (*Toxoplasma gondii*) | CPS | PMID:11859373 **Fox and Bzik, 2002** | | |
| Antibody | *Toxoplasma gondii* Rabbit polyclonal | Collaborator | | IHC: 1:100 |
| Antibody | iNOS Rabbit polyclonal | Abcam | Cat # ab15323, RRID:AB_301857 | IHC 1:50 |
| Antibody | ERTR7 Af647 Rat monoclonal (sc-73355) | Santa Cruz Biotechnology | Cat # sc-73355 RRID:AB_1122890 | IF (1:50) |
| Antibody | F4/80 BV480 Rat monoclonal (T45-2342) | BD | Cat # 565635 RRID:AB_2739313 | IF (1:25) |
| Antibody | CD45 Af700 Rat monoclonal (30-F11) | BioLegend | Cat # 103127, RRID:AB_493714 | IF (1:20) |
| Antibody | CD335 NKp46 PE/Dazzle 594 Rat monoclonal (29A1.4) | BioLegend | Cat # 137629, RRID:AB_2616665 | FC (1:200) |
| Antibody | NK-1.1 BV711 Mouse monoclonal (PK136) | BioLegend | Cat # 108745, RRID:AB_2563286 | FC (1:200) |
| Antibody | IFN gamma Af700 Rat monoclonal (XMG1.2) | Thermo Fisher | Cat # 56-7311-82, RRID:AB_2688063 | FC (1:200) |
| Antibody | CD200 Receptor APC Rat monoclonal (OX110) | Thermo Fisher | Cat # 17-5201-82, RRID:AB_10717289 | FC (1:200) |
| Antibody | T1/ST2 Biotin Rat monoclonal (DJ8) | MD Biosciences | Cat # 101001B, RRID:AB_947551 | FC (1:200) |
| Antibody | T-bet PE-Cy7 Mouse monoclonal (4B10) | BioLegend | Cat # 644823 | FC (1:200) |
| Antibody | EOMES PE Rat monoclonal (Dan11mag) | Thermo Fisher | Cat # 12-4875-82, RRID:AB_1603275 | FC (1:200) |
| Antibody | CD11b ef450 Rat monoclonal (M1/70) | Thermo Fisher | Cat # 48-0112-80, RRID:AB_1582237 | FC (1:1000) |
| Antibody | CD11c APC-ef780 Armenian hamster monoclonal (N418) | Thermo Fisher | Cat# 47-0114-80, RRID:AB_1548663 | FC (1:200) |
| Antibody | Ly-6C BV785 Rat monoclonal (HK1.4) | BioLegend | Cat # 128041, RRID:AB_2565852 | FC (1:200) |
| Antibody | Ly-6G BV711 Rat monoclonal (1A8) | BioLegend | Cat # 127643, RRID:AB_2565971 | FC (1:200) |
| Antibody | CCR2 CD192 APC Rat monoclonal (SA203G11) | BioLegend | Cat # 150628, RRID:AB_2810415 | FC (1:200) |

*Continued on next page*

*Continued*

| Reagent type (species) or resource | Designation | Source or reference | Identifiers | Additional information |
|---|---|---|---|---|
| Antibody | CD64 FcgammaRI PE-Cy7 Mouse monoclonal (X54-5/7.1) | BioLegend | Cat # 139306, RRID:AB_11219391 | FC (1:200) |
| Antibody | MHC Class II (I-A/I-E) Af700 Rat monoclonal (M5/114.15.2) | Thermo Fisher | Cat # 56-5321-82, RRID:AB_494009 | FC (1:200) |
| Antibody | iNOS APC Rat monoclonal (CXNFT) | Thermo Fisher | Cat # 17-5920-82, RRID:AB_2573244 | FC (1:200) |
| Antibody | Podoplanin gp38 PerCP-ef710 Syrian hamster monoclonal (eBio8.1.1) | Thermo Fisher | Cat # 46-5381-82, RRID:AB_2848339 | FC (1:200) |
| Antibody | CD31 BV605 Rat monoclonal (390) | BioLegend | Cat # 102427, RRID:AB_2563982 | FC (1:200) |
| Peptide, recombinant protein | Recombinant murine IL-33 | Peprotech | Cat # 210–33 | |
| Commercial assay or kit | IL-33 ELISA | R and D Biosystems | Cat # DY3626 | |

## Mice

B6 (C57BL/6NTac) (Taconic #B6-F), $Rag1^{-/-}$ (B6.129S7-$Rag1^{tm1Mom}$/J) (Jackson #002216), and $Rag2^{-/-}$; $Il2rg^{-/-}$(C57BL/6NTac.$Rag2^{tm1Fwa}$;$Il2rg^{tm1Wjl}$) (Taconic #4111) mice were purchased from their respective vendors. $Il33^{-/-}$ ($Il33^{tm1.1Arte}$) (Jackson #350163) mice were provided by MedImmune (now AstraZeneca). $Il33^{fl/fl}$-eGFP (B6(129S4)-Il33$^{tm1.1Bryc}$/J), originally generated by Paul Bryce, were obtained locally from Dr. De'Broski Herbert. IL-33R KO ($Il1rl1^{-/-}$) mice, originally derived by Andrew McKenzie (*Townsend et al., 2000*) (University of Cambridge) and back-crossed to C57BL/6 by Peter Nigrovic (Harvard University), were provided by Edward Behrens at Children's Hospital of Philadelphia. $Rag1^{-/-}$; $Il1rl1^{-/-}$ mice were generated by crossing the knockouts described above. Analysis of these uninfected KO mice revealed no obvious developmental defects, while their immune compartments appeared comparable to $Rag1^{-/-}$; $Il1rl1^{+/+}$ mice in cell numbers and phenotype at homeostasis. Mice were housed in a specific pathogen-free environment at the University of Pennsylvania School of Veterinary Medicine and treated according to protocols approved by the Institutional Animal Care and Use Committee at the University. Male and female (age 8–12 weeks at start of experiment) mice were used for all experiments.

## Parasites and infection

The ME49 strain of *T. gondii* was maintained by serial passage in Swiss Webster mice and used to generate banks of chronically infected CBA/ca mice, which were a source of tissue cysts for these experiments. Pru-derived transgenic parasites and CPS parasites were maintained in cultured human fibroblasts in DMEM supplemented with 10% FBS. For CPS parasites, supplemental uracil was also added to media. For all experiments presented here, mice were infected intraperitoneally (i.p.) with 20 cysts (ME49), or $1 \times 10^4$ tachyzoites (Pru), or $2 \times 10^5$ tachyzoites (CPS). Soluble toxoplasma antigen was prepared from tachyzoites of the RH strain as described previously (*Hauser et al., 1983*). For quantitative PCR (qPCR), DNA was isolated from tissues using the DNEasy DNA isolation kit (Qiagen) followed by qPCR measuring the abundance of the *T. gondii* gene B1 using the primers 5'-TCTTTAAAGCGTTCGTGGTC-3' (forward) and 5'-GGAACTGCATCCGTTCATGAG-3' (reverse).

## Histology

For IHC detection of *T. gondii* and iNOS, tissues were fixed in 10% formalin solution and then paraffin embedded and sectioned. Sections were deparaffinized, rehydrated, Ag retrieved in 0.01 M sodium citrate buffer (pH 6.0), and endogenous peroxidase blocked by 0.3% $H_2O_2$ in PBS. After

blocking with 2% normal goat serum, the sections were incubated either with rabbit anti-Toxo-plasma Ab, anti-INOS Ab, or isotype control. The sections were then incubated with biotinylated goat anti-rabbit IgG (Vector, Burlingame, CA), and ABC reagent was applied (Vectastain ABC Kit; Vector Labs). Then DAB substrate (Vector Labs) was used to visualize specific staining according to manufacturer's instructions, and slides were counterstained with hematoxylin. To quantify parasite burden in the peritoneal exudate, 100,000 cells were used to prepare cytospins. Cells were methanol fixed and then stained with the Protocol Hema-3 Stain Set, and the ratio of infected cells to total cells in a field of view was calculated, with a minimum of 200 cells counted per sample. For whole tis-sue mount immunofluorescence staining, omenta were harvested from mice and fixed in 1% PFA overnight at 4°C. After rinsing, tissue was blocked using 10% bovine serum albumin (BSA), 0.5% nor-mal rat serum (Invitrogen), and 1 µg/ml 2.4G2 (BD) in PBS for 1 hr at room temperature. Omenta were next incubated in PBS containing primary antibodies at 4°C for 3 days and subsequently rinsed with PBS overnight. Antibodies used for this analysis: ERTR7 (sc-73355, Santa Cruz Biotechnology), F4/80 (T45-2342, BD), and CD45 (30-F11, Biolegend).

Immunofluorescence combining IL-33 (R and D AF3626) and CD45 (Biolegend 30-F11) antibodies was performed using the OPAL Automation Multiplex IHC Detection Kit (Akoya Biosciences, Cat # 160 #NEL830001KT) implemented onto a BOND Research Detection System (DS9455). All wide-field images were obtained on a Leica DM6000 microscope using the Leica Imaging Suite software. Confocal images were acquired on a Leica STED 3× Super-resolution microscope. Image analysis was performed using FIJI and Imaris software packages.

## Generation of lymphokine-activated killer cells

Lymphokine-activated killer cells (LAKs) were generated from $Rag1^{-/-}$ bone marrow as described previously (*Hunter et al., 1997*; *Wherry et al., 1991*). Briefly, whole bone marrow was plated at 1 M cells/ml in cRPMI +400 U/ml Proleukin human IL-2 (Peprotech). Fresh IL-2 was added every third day, and cells were used for experiments between days 7 and 10.

## Antibody and cytokine reagents

For in vitro assays, recombinant IL-33 was purchased from Peprotech (Cat # 210–33 Rocky Hill, NJ). For in vivo treatment experiments, recombinant IL-33 (MedImmune), which was modified to be resis-tant to oxidation, was used, as described previously (*Cohen et al., 2015*). IL-33 DuoSet ELISA was purchased from R and D Biosystems (Cat # DY3626, Minneapolis, MN). For flow cytometry, the fol-lowing combinations of antibodies were used: for analysis of NK cells: CD335 NKp46 (29A1.4, eBio-science), NK-1.1 (PK136, Biolegend), IFN-γ (XMG1.2, eBioscience), CD200R1 (OX110, eBioscience), IL-33R (DJ8, MD Biosciences), T-bet (4B10, Biolegend), and EOMES (Dan11mag, eBioscience). For analysis of myeloid cells: CD11b (M1/70, eBioscience), CD11c (N418, Biolegend), Ly6c (HK1.4, Biole-gend), Ly6g (1A8, Biolegend), CCR2 CD192 (SA203G11, Biolegend), CD64 FcgRI (X54-5/7.1, Biole-gend), MHC II I-A/I-E (m5/114.15.2, eBioscience), iNOS (CXNFT, eBioscience), IL-33R (DJ8, MD Biosciences). For analysis of stromal cells: gp38/PDPN (8.1.1, Biolegend), CD31 (390, Biolegend). Flow cytometry was performed on BD Fortessa and X-50 cytometers, and data analysis was per-formed using Flowjo nine and Flowjo 10 (Treestar), and Prism 7 and 8 (Graphpad). Uniform Manifold Approximation and Projection for Dimension Reduction (uMAP) analysis was performed using the uMAP plug-in (version: 1802.03426, 2018, 2017, Leland McInnes) for Flowjo (Version 10.53). The Euclidean distance function was utilized with a nearest neighbor score of 15 and a minimum distance rating of 0.5.

## Quantification and statistical analysis

All data are expressed as means ± standard error of the mean (SEM). For comparisons between two groups, the Student's t-test was applied. For data with more than two data sets, one-way ANOVA coupled with Tukey's multiple comparisons test was applied. Statistical details are indicated in figure legends.

## Acknowledgements

The authors would like to acknowledge the contributions of the Penn Vet Imaging Core and the Comparative Pathology Core at the University of Pennsylvania School of Veterinary Medicine, and

the members of the Hunter laboratory for scientific and moral support. The model in *Figure 6* was generated using assets from BioRender.com. This work was supported by grants from the National Institutes of Health: NIAID 5R01AI125563 and 5T32-AI007532-21

## Additional information

### Competing interests

Jonathan Silver: is a full-time employee and shareholder of AstraZeneca. The other authors declare that no competing interests exist.

### Funding

| Funder | Grant reference number | Author |
|---|---|---|
| National Institute of Allergy and Infectious Diseases | 5R01AI125563-05 | Christopher A Hunter |
| National Institute of Allergy and Infectious Diseases | 5T32AI00753223 | Christopher A Hunter |

The funders had no role in study design, data collection and interpretation, or the decision to submit the work for publication.

### Author contributions

Joseph T Clark, Conceptualization, Data curation, Formal analysis, Validation, Investigation, Visualization, Methodology, Writing - original draft, Project administration, Writing - review and editing; David A Christian, Resources, Software, Investigation, Visualization, Methodology, Writing - review and editing; Jodi A Gullicksrud, Conceptualization, Resources, Investigation, Methodology, Writing - review and editing; Joseph A Perry, Software, Formal analysis, Investigation, Methodology; Jeongho Park, Resources, Validation, Investigation, Methodology; Maxime Jacquet, Validation, Investigation, Methodology; James C Tarrant, Resources, Investigation, Methodology, Writing - review and editing; Enrico Radaelli, Resources, Methodology, Writing - review and editing; Jonathan Silver, Resources, Writing - review and editing; Christopher A Hunter, Conceptualization, Supervision, Funding acquisition, Methodology, Writing - original draft, Project administration, Writing - review and editing

### Author ORCIDs

Joseph T Clark https://orcid.org/0000-0001-7764-6000
Christopher A Hunter https://orcid.org/0000-0003-3092-1428

### Ethics

Animal experimentation: This study was performed in strict accordance with the recommendations in the Guide for the Care and Use of Laboratory Animals of the National Institutes of Health. All of the animals were handled according to approved institutional animal care and use committee (IACUC) protocols (#805045) of the University of Pennsylvania.

### Decision letter and Author response

Decision letter https://doi.org/10.7554/eLife.65614.sa1
Author response https://doi.org/10.7554/eLife.65614.sa2

## Additional files

### Supplementary files

• Transparent reporting form

## Data availability

All data generated or analysed during this study are included in the manuscript and supporting files.

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
