## [Decision Letter]

**Acceptance summary:**

Thank you for your comprehensive revisions. These changes greatly strengthen the work and provide insight to how IL-33 regulates ILC functions.

**Decision letter after peer review:**

Thank you for submitting your article "IL-33 promotes innate lymphoid cell-dependent IFN-γ production required for innate immunity to *Toxoplasma gondii*" for consideration by *eLife*. Your article has been reviewed by 3 peer reviewers, one of whom is a member of our Board of Reviewing Editors, and the evaluation has been overseen by Carla Rothlin as the Senior Editor. The following individual involved in review of your submission has agreed to reveal their identity: Malcolm R Starkey (Reviewer #3).

Essential Revisions:

1. Delineate more clearly the roles of the different compartments and mechanistic interactions.

2. Significantly improve the technical reporting of how experiments were performed in the study. eg. not clear how infections were performed; not clear which data is referred to in the text; were controls used.

3. Clarify the models used especially the use of controls and IL-33 reporter mice.

*Reviewer #1 (Recommendations for the authors):*

Figure 1. Although Rag^-/-^ mice were examined in Figure 1A, which mice the data report for the remainder of the figure has not been made clear. It is assumed that it is wild-type mice. What is the situation in the Rag^-/-^ mice.

Figures: It would be helpful for the authors to clarify exactly how many experiments are shown in each figure. While they say "Data are representative of" it is often unclear whether they are actually showing pooled data, or one experiment representative of X similar independent experiments.

This work would be strengthened significantly by delineating more clearly the contributions of each compartment. This is a major confusion throughout the manuscript in trying to assess which cells are actually important in this response rather than a catalogue of cells that produce IL-33 or respond to IL-12. Thus, the causal relationships have not bee defined.

This work would benefit from a schematic to indicate how the authors believe the different cells are connected and which are the real drivers/where connections have been demonstrated in driving the immune response.

*Reviewer #2 (Recommendations for the authors):*

1. The authors appear to "force for enhance" a role for IL-33 in this experimental model by injecting anti-IFN-γ antibody to the mice. Without such antibody treatment, there was undetectable level of IL-33 (Figure 1A and B) following infection of both WT and Rag deficient mice. This seriously questions the role of IL-33 in immunity. The authors should incorporate this critical concept in their discussion and emphasize that these role of ILL-33 is limited to situations where IFN-g response is suboptimal or defective.

2. It is not clear how the authors have performed infection. The only detail about the infection protocol is only explained for Figure 1A. Figure legends for other figures should include details of the infection procedure.

3. The naïve IL-33 GFP reporter mouse shows higher percentage of IL-33 compared to infected IL-33 GFP reporter mice (Figure 1C),, which is not in agreement with figures 1A and B. Given that naive mice are reporting IL-33 at about 19 times that of infected controls (albeit not coming CD5^+^ cells), one wonders whether this reporter mice is really ideal for assessing the role of IL-33 during infection.

4. The authors should check for the role of IL-33 in neutrophil migration and expansion during early *T. gondii* infection.

5. Is f there any change (increase) in ILC1 and NK cells numbers in RAG^-/-^ mice treated with rIL-33.

*Reviewer #3 (Recommendations for the authors):*

The manuscript by Clark et al., is well written, generally well presented and approached with scientific rigour in terms of experimental design and approach. Several comments may assist with the communication of the results to the target audience.

1. Abbreviations at first mention. Many abbreviations are used without explanation at first mention. This helps no experts follow the story. For example in introduction IL, NK, iNOS, IFN, TH2 etc. Please address all throughout where appropriate.

2. Some data panels are presented as bar graphs rather than individual points. Preference would be for all data to be presented as individual points so that the reader can get a rapid appreciation of the data, sample size etc.

3. Figure 1G. It appears in the image that the CD45+ staining has not worked optimally? Perhaps it is the image quality, but appears to be some non specific staining

4. For FACS plots, could pseudocolour plots be used? These are standard in field and easier to visualise

5. Terminology with gc^-/-^. In some parts is gc^-/-^ in others IL2Rg^-/-^. Consistency would help here.

6. Were any studies with anti-IL-33 in infected wild-type mice performed? If not could this at least be discussed? Whilst knockout mouse models are valuable, the indirect impact of cell/factor deficiency throughout development should be considered. May disrupt immune cell development, microbiome etc.

---

## [Author Response]

Essential Revisions:1. Delineate more clearly the roles of the different compartments and mechanistic interactions.

**Our data support a paradigm in which in which infection with *T. gondii* stimulates release of IL-12 by dendritic cells but the release of IL-33 from infected stromal cells. IL-12 induces innate lymphoid cells to express the IL-33 receptor and the combination of signals from IL-12 and IL-33 stimulate ILC expansion and the production of IFN-γ. In turn, IFN-γ is responsible for the activation of monocytes that are the effector population that limits the growth of *T. gondii*. The text and figures have been updated to emphasize this paradigm, including new data regarding levels of IL-33R expression by CD45+ cells (Figure 1 Supplement 1E). As suggested, a model that depicts these events has been. added to the submission (Figure 6).**

2. Significantly improve the technical reporting of how experiments were performed in the study. eg. not clear how infections were performed; not clear which data is referred to in the text; were controls used.

**All figures have been converted to individual data points, and all figure legends have been amended to clarify that all data shown are from single experiments and representative of a specified number of replicates. The text has been updated to clarify methods and details for each experiment, including that all infections were performed i.p. (e.g., lines 136, 197, 218, 276, 295, and all figure legends). Control data in the form of knockout mice has been added to experiments as appropriate, including the addition of analysis of different tissues from IL-33^-/-^ mice to Figure 1 Supplement 1 as validation of the reporter model.**

3. Clarify the models used especially the use of controls and IL-33 reporter mice.

**A discussion of the validation of the IL-33 reporter mouse has been added to the text (line 217) and knockout mouse controls for the IL-33 reporters has been adding to Figure 1 Supplement 1. In addition, details of the characterization of the Rag^-/-^ IL-33R^-/-^ mouse has been added to the methods section (line 125) that details the lack of overt immunological defects in the development of these mice.**

Reviewer #1 (Recommendations for the authors):Figure 1. Although Rag^-/-^ mice were examined in Figure 1A, which mice the data report for the remainder of the figure has not been made clear. It is assumed that it is wild-type mice. What is the situation in the Rag^-/-^ mice.

**In Figure 1 we use a combination of immune competent IL-33 reporter mice and ELISA to look at infection-induced changes in IL-33 levels. The reviewer is correct that the reporters were only used for WT mice as we do not have the IL-33 reporter crossed to a Rag^-/-^ background. However, by ELISA, the levels of IL-33 in WT and Rag mice are broadly comparable (see Figure 1A) and track with levels of parasite replication.**

Figures: It would be helpful for the authors to clarify exactly how many experiments are shown in each figure. While they say "Data are representative of" it is often unclear whether they are actually showing pooled data, or one experiment representative of X similar independent experiments.

**All figures have been converted to show individual data points and figure legends amended to reflect this. All figures are data sets from individual experiments representative of multiple replicates.**

This work would be strengthened significantly by delineating more clearly the contributions of each compartment. This is a major confusion throughout the manuscript in trying to assess which cells are actually important in this response rather than a catalogue of cells that produce IL-33 or respond to IL-12. Thus, the causal relationships have not been defined.

**The data indicate that in vivo non-hematopoietic cells are the major source of IL-33 and that IL-12 is important for its ability to induce expression of IL-33R on ILC1 and NK cells. This point has been clarified in the Discussion section of the revised submission (line 262) and is now highlighted in the model shown in Figure 6.**

This work would benefit from a schematic to indicate how the authors believe the different cells are connected and which are the real drivers/where connections have been demonstrated in driving the immune response.

**A model has been included in the revised submission as Figure 6.**

Reviewer #2 (Recommendations for the authors):1. The authors appear to "force for enhance" a role for IL-33 in this experimental model by injecting anti-IFN-γ antibody to the mice. Without such antibody treatment, there was undetectable level of IL-33 (Figure 1A and B) following infection of both WT and Rag deficient mice. This seriously questions the role of IL-33 in immunity. The authors should incorporate this critical concept in their discussion and emphasize that these role of ILL-33 is limited to situations where IFN-g response is suboptimal or defective.

**Perhaps there has been a mis-understanding. Our data do not support the conclusion that the role of IL-33 is limited to situations where the IFN-g responses is sub-optimal or defective. It is important to note that there are elevated levels of IL-33 in Rag^-/-^ mice, even without treatment with anti-IFN-g (Figure 1A). Additional data sets show the increased levels of IL-33^+^ cells in WT mice without treatment with anti-IFN-g. The use of anti-IFN-g, and the use of the CPS mutants (that do not replicate) establish the link between amount of parasite replication and levels of IL-33. Moreover, the data that Rag^-/-^ mice deficient in IL-33R show elevated parasite burdens and diminished ILC responses supports a physiological role of endogenous IL-33 in innate resistance to *T. gondii*. In addition, we now include data sets that (Rag sufficient) IL-33 KO have reduced levels of IFN-γ and increased weight loss at 10 dpi and elevated parasite burdens prior to entering the chronic stage of infection. These data have been added to the revised manuscript as Figure 3 Supplement 1 A and B and discussed on lines 283-284. Lastly, previous studies, that we have cited, have shown that these mice have an increased susceptibility to the chronic stage of infection.**

2. It is not clear how the authors have performed infection. The only detail about the infection protocol is only explained for Figure 1A. Figure legends for other figures should include details of the infection procedure.

**The revised text has been updated to indicate that all infections are intraperitoneal: line #s 136, 197, 218, 276, 295, and all figure legends.**

3. The naïve IL-33 GFP reporter mouse shows higher percentage of IL-33 compared to infected IL-33 GFP reporter mice (Figure 1C),, which is not in agreement with figures 1A and B. Given that naive mice are reporting IL-33 at about 19 times that of infected controls (albeit not coming CD5^+^ cells), one wonders whether this reporter mice is really ideal for assessing the role of IL-33 during infection.

**Firstly, we do validate the use of the IL-33 reporter in Figure 1 Supplement 1 and show the fidelity between reporter and protein expression. The reviewer appears concerned about the changes in the percentage of IL-33^+^ stromal cells. The decreased frequency of IL-33^+^ cells observed in the omenta of infected mice is a consequence of the massive influx of CD45^+^ cells upon infection. These data indicate that the majority of IL-33 expression is associated with CD45^-^ cells. In addition, the overall number of IL-33^+^ cells and the overall level of IL-33-GFP signal increases as the size of the milky spots increases, as is shown in the subsequent panels.**

4. The authors should check for the role of IL-33 in neutrophil migration and expansion during early T. gondii infection.

**We appreciate this suggestion. The revised manuscript includes Figure 3 Supplement 1C showing that IL-33R deficiency was associated with a 2-fold increase in the number of neutrophils in the peritoneum relative to WT mice. We now note that enhanced neutrophil responses are typically associated with increased levels of parasite replication. IL-33 treatment was not associated with any change in neutrophil numbers (data not shown). Discussion of these data is integrated into the Results section (lines 290-291, 303-304).**

5. Is f there any change (increase) in ILC1 and NK cells numbers in RAG^-/-^ mice treated with rIL-33.

**Thank you for this suggestion. There was an increase in ILC1 and NK cell numbers in IL-33-treated Rag^-/-^ mice relative to mock treated mice. These data are now included in Figure 5 of the revised manuscript and are discussed on lines 322-323.**

Reviewer #3 (Recommendations for the authors):The manuscript by Clark et al., is well written, generally well presented and approached with scientific rigour in terms of experimental design and approach. Several comments may assist with the communication of the results to the target audience.1. Abbreviations at first mention. Many abbreviations are used without explanation at first mention. This helps no experts follow the story. For example in introduction IL, NK, iNOS, IFN, TH2 etc. Please address all throughout where appropriate.

An abbreviation key has been added to the revised manuscript and key abbreviations defined on first mention.

2. Some data panels are presented as bar graphs rather than individual points. Preference would be for all data to be presented as individual points so that the reader can get a rapid appreciation of the data, sample size etc.

**Figures 1, 2, 3, 4, and 5 have been converted to individual data points.**

3. Figure 1G. It appears in the image that the CD45+ staining has not worked optimally? Perhaps it is the image quality, but appears to be some non specific stainingThank you for the suggestion. CD45 display was indeed sub-optimal, it was adjusted to display dimly due to the abundance of CD45+ cells, so as to not overwhelm the merged image. The brightness of CD45 has been increased slightly in the revised Figure 1G.*4. For FACS plots, could pseudocolour plots be used? These are standard in field and easier to visualise*

**We appreciate this suggestion and typically use a combination of plots and only use pseudocolor when there is a specific benefit, eg see Figure 1D. For the majority of our plots we prefer contour plots including outliers in part because they do not require color.**

5. Terminology with gc^-/-^. In some parts is gc^-/-^ in others IL2Rg^-/-^. Consistency would help here.

**This inconsistency has been fixed.**

6. Were any studies with anti-IL-33 in infected wild-type mice performed? If not could this at least be discussed? Whilst knockout mouse models are valuable, the indirect impact of cell/factor deficiency throughout development should be considered. May disrupt immune cell development, microbiome etc.

**Firstly, no gross defects in Rag^-/-^ IL33R^-/-^ mice were observed at homeostasis. It should also be noted that (with the exception of *T. gondii*) IL-33 deficient mice are remarkably resistant to other infections suggesting there is no major immune deficiency in mice. The text has been revised to discuss this point (lines 125-127). We did not perform any studies inhibiting IL-33, as there is no commercially available IL-33 blocking antibody but we do now present data with Rag-sufficient IL-33 mice. In addition, the protective effects of IL-33 treatment on the ILC compartment are consistent with how the loss of IL-33 in Rag^-/-^ mice leads to reduced ILC responses.**